# Bridging the Sim-to-Real Gap for Efficient and Robust Robotic Skill Acquisition

**Jingzhi Cui**
IIIS, Tsinghua University
cjz24@mails.tsinghua.edu.cn

**Yutao Ouyang**
IIIS, Tsinghua University
oyyt24@mails.tsinghua.edu.cn

## Abstract

In recent years, the integration of learning-based methods, particularly Imitation Learning (IL) and Reinforcement Learning (RL), has brought significant advancements to robotics applications. These methods enable robots to learn complex behaviors, with IL facilitating rapid skill acquisition via expert demonstrations and RL permitting self-discovery of optimal strategies. However, the reliance on real-world data presents substantial challenges, as data collection can be resource-intensive, time-consuming, and risky for robotic systems. While simulation environments have emerged as a practical solution, providing abundant training data, they also introduce the sim-to-real gap—a critical challenge that hampers the effective transfer of learned behaviors from simulations to real-world scenarios due to discrepancies in sensor performance, environmental conditions, and material properties.

To address these challenges, we propose a novel framework that combines the strengths of both IL and RL while minimizing transfer difficulties associated with simulation-trained models. Our approach leverages the cost-effectiveness of simulated data to enhance robot learning outcomes, utilizing advanced techniques to improve transferability and reduce the sim-to-real gap. By harmonizing the efficiency of IL with the autonomy of RL, we aim to create a more effective learning paradigm that not only accelerates skill acquisition but also enhances real-world applicability, which may pave the way for more autonomous and capable robotic systems.

## 1 Background

In recent years, learning-based methods like Imitation Learning (IL) and Reinforcement Learning (RL) have shown promising success in robotics applications. These approaches allow robots to learn complex behaviors by interacting with environments and, in the case of IL, by leveraging expert demonstrations. IL enables robots to quickly acquire skills through observation, while RL allows them to autonomously discover strategies through trial and error guided by reward signals. Together, these methods offer a framework for end-to-end learning, from perception to control, that has opened new avenues in robot autonomy.

However, most learning-based methods heavily rely on real-world data, which is often expensive and challenging to collect. Real-world data acquisition requires substantial resources, including time and equipment, and risks potential damage to the robot in early training stages. To address these limitations, researchers increasingly use simulation environments, which can provide abundant and safe data for training through parallelized trials. Yet, simulation-based training introduces the sim-to-real gap: unavoidable discrepancies between simulated and real-world conditions—such as differences in sensor accuracy, lighting, and physical properties—can hinder a model's real-world performance. This sim-to-real gap remains a critical challenge, making it essential to develop approaches that improve the transferability of simulation-trained models to actual robotic tasks.

Preprint. Under review.

In summary, while RL and IL each offer distinct advantages, RL's autonomy and IL's guided efficiency must be combined to enhance learning outcomes. Simulation environments present a viable pathway for scalable training, but closing the sim-to-real gap is essential to fully utilize simulated data in real-world contexts. Our work addresses these two objectives: (1) minimizing transfer difficulty to leverage the cost-effectiveness of simulated data, and (2) combining RL and IL to create a more efficient learning approach.

## 2 Related Work

### 2.1 Reinforcement Learning

Reinforcement Learning (RL) acquires strategies through interaction with the environment, typically using a neural network policy that maps observations to actions. Simulation environments improve RL efficiency by safely modeling varied scenarios, enabling more robust strategies without real-world testing risks.In 2019, Hwuangbo et al. introduced a Trust Region Policy Optimization (TRPO)[1] approach for control policies[2]. Lee et al. (2020) proposed a teacher-student framework[3] where a teacher policy uses privileged information to guide a student's learning under limited observation. In 2021, Rudin et al. developed an environment[4] simulating thousands of robots on a single GPU, achieving deployable policies within 20 minutes using Proximal Policy Optimization (PPO). They further introduced an end-to-end local navigation policy for challenging terrains[5], addressing traditional control limitations with a flexible reward formulation based on goal positions.

### 2.2 Imitation Learning

Imitation Learning (IL) enables robots to acquire skills from expert demonstrations, reducing trial and error and increasing safety in training. However, collecting sufficient demonstrations can be costly, and learned policies often lack robustness in new scenarios.In 2018, Ho and Ermon introduced Generative Adversarial Imitation Learning (GAIL), which enables agents to mimic expert behavior without explicit reward functions[6]. Zhang et al. improved IL robustness by adding noise and data augmentation[7]. Torabi et al. introduced Behavioral Cloning from Observation (BCO), which trains agents from state transitions alone, reducing data requirements[8]. In 2021, Rafailov et al. proposed a meta-IL framework that adapts policies more efficiently to new tasks by learning from distributions of demonstrations[9]. These methods highlight IL's progress toward more adaptable and data-efficient learning.

### 2.3 Sim to Real

In robotics, several approaches aim to bridge the gap between simulation and real-world environments. Domain Randomization introduces randomness in simulation (e.g., lighting, object textures) to approximate real-world diversity, though this increases the model's learning complexity by requiring it to filter out noise[10]. Domain Adaptation focuses on feature or pixel-level transformation to make simulated images resemble real-world scenes, typically requiring large amounts of real-world data[11][12]. Conversely, Real-to-Sim Transfer adapts real-world inputs into simulation-style formats, allowing simulation-trained policies to apply directly in the real world; however, this approach depends on auxiliary data like depth maps, and GAN-based training can be unstable.

## 3 Proposed Method

This study investigates how to train robots exclusively with RGB images as observations, using only efficient, low-cost simulation data, and without any real-world data to simplify the transfer to real-world applications. The objective is to maximize transferability while achieving high success rates in real-world robotic tasks. To this end, we first translates real-world images into a standardized simulation-style, ensuring consistency across different visual domains. Next, within the simulation environment, we develop a robust grasping policy by combining imitation learning and reinforcement learning techniques. This dual approach enables the trained policy to be directly deployed in real-world environments, maintaining high performance despite the absence of real-world training data.

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
