# OpenReview forum: "Bridging the Sim-to-Real Gap for Efficient and Robust Robotic Skill Acquisition"
_tsinghua.edu.cn/THU/2024/Fall/AML — THU 2024 Fall AML Submission_

### Official Review · ~Shaoting_Zhu1 · 2024-11-06
**Review of submission 45**

**Rating:** 8
**Confidence:** 4

**Review:**

This proposal presents a novel framework that combines Imitation Learning (IL) and Reinforcement Learning (RL) to address the challenges of transferring robotic skills from simulated environments to real-world scenarios. The authors highlight the sim-to-real gap as a critical issue in robotics, where discrepancies between simulation and real-world conditions can impede the effectiveness of learned behaviors.

**Stength**
1. Innovative Integration of IL and RL: The paper proposes a unique approach that leverages the rapid skill acquisition capabilities of IL and the self-discovery of optimal strategies through RL. This dual approach aims to create a more efficient learning paradigm that enhances both skill acquisition and real-world applicability.
2. Addressing the Sim-to-Real Gap: The authors focus on minimizing the transfer difficulties associated with simulation-trained models, which is a significant contribution to the field of robotics, given the high stakes and risks involved in real-world robotic operations.

**Weakness**
1. Lack of detailed method: The proposed method outlined in this article is rather brief and general, and it does not provide a clear view of the detailed solutions.
2. Tasks and objectives are not clearly defined: The proposal does not specify the type of robot morphology or specific robotic tasks to which the method will be applied, nor does it present clear metrics.

---

### Official Review · ~Chua_Shei_Pern1 · 2024-11-06
**Good**

**Rating:** 9
**Confidence:** 4

**Review:**

The proposal effectively outlines a hybrid learning approach to bridge the sim-to-real gap in robotic skill acquisition. While the proposed work seems promising, further clarity on specific metrics for evaluating transfer success would enhance the proposal's grounding.

---

### Official Review · ~Lily_Sheng1 · 2024-11-08
**Submission 45 Review**

**Rating:** 8
**Confidence:** 4

**Review:**

This work explores a method for training robotic systems using a combination of Reinforcement Learning and Imitation Learning to transfer learned policies from simulation to real-world applications. The approach focuses on maximizing transferability by translating real-world images to a consistent simulation style and using RGB-only observations, which are then used to develop a robust grasping policy.

Pros:
1. The exclusive use of simulated data reduces the cost of training with real-world data and real-world data collection.

Cons:
1. Relying completely on simulated RGB data may not be able to capture the entire complexity of real-world environments.
2. There is little detail on how the grasping policy will be tested and applied in a real-world world environment.

---

### Official Review · ~Ziang_Zheng1 · 2024-11-09
**a promising approach to bridging the sim-to-real gap for robotic skill acquisition**

**Rating:** 8
**Confidence:** 3

**Review:**

**Reviewer Response**

The paper presents a promising approach to bridging the sim-to-real gap for robotic skill acquisition using a framework that combines Imitation Learning (IL) and Reinforcement Learning (RL) to achieve efficient, robust skill transfer from simulation to the real world. By exclusively using RGB images and simulated data, the authors address the challenges of resource-intense, risky real-world training and aim to maximize transferability and real-world performance. The approach is well-motivated and relevant, especially given the need for scalable, safe training methods in robotics.

### Strengths

1. **Clear Problem Identification**: The paper effectively outlines the challenges of sim-to-real transfer and highlights the complementary strengths of IL and RL in addressing these issues.

2. **Use of Simulation-Only Data**: Focusing exclusively on simulation-trained models without any real-world data presents a cost-effective and safe training paradigm that could be impactful for many real-world applications. This approach is particularly beneficial for tasks with high real-world setup costs or operational risks.

3. **Integrative Methodology**: The integration of IL for rapid skill acquisition with RL’s robustness in unstructured environments is well-conceived. The emphasis on domain standardization across visual inputs to reduce discrepancies between simulation and real-world conditions shows strong insight into sim-to-real challenges.

### Areas for Improvement

1. **Experimental Results and Real-World Validation**: The paper would benefit significantly from real-world validation of the proposed framework’s effectiveness. Providing quantitative metrics for success rates, transfer efficiency, and any limitations observed in preliminary tests would strengthen the overall contribution.

2. **Comparative Analysis with Existing Methods**: Although the paper mentions domain randomization, adaptation, and real-to-sim techniques, it lacks a detailed comparison of the proposed method's performance with these established techniques. A side-by-side evaluation could highlight the unique advantages of the proposed approach.

3. **Specifics on Visual Domain Standardization**: The process of translating real-world images into a standardized simulation style for consistency is an interesting and novel component. However, more details on the technical aspects of this transformation and its impact on transfer performance would enhance the clarity and rigor of the paper.

4. **Adaptability and Scalability**: The framework's adaptability to various robotic tasks beyond grasping, as well as its scalability to multi-robot setups, would be valuable areas to address. Providing insights into how adaptable and scalable this framework is would increase its relevance to a broader audience.

5. **Handling of Noise and Variability in Simulated Environments**: Although domain standardization aims to address sim-to-real discrepancies, it is important to discuss the handling of potential noise and variability within the simulated environments themselves. Clarifying this could provide insights into the robustness of the trained policies.

### Recommendation

The paper provides a solid foundation for bridging the sim-to-real gap through a simulation-based, IL-RL framework. Addressing the above concerns—particularly around experimental validation and comparative analysis—would strengthen the contribution and applicability of the framework. I recommend acceptance with major revisions, especially with an emphasis on real-world performance metrics and detailed technical explanations of the standardization approach.

**Suggested Action**: Accept with Major Revisions

---

### Official Review · ~Lei_Wu17 · 2024-11-09
**Evaluation of "Bridging the Sim-to-Real Gap for Efficient and Robust Robotic Skill Acquisition"**

**Rating:** 7
**Confidence:** 4

**Review:**

# Pros
* Innovative Approach: The dual-method approach using IL and RL with RGB-only observations could simplify and expedite robot training.
* Cost-Effective: By minimizing real-world data requirements, the framework could lower the cost of training robotic systems.
* Comprehensive Related Work: The paper provides a solid overview of existing methods, framing its approach within current research trends.
* Potential Impact: Addresses a crucial bottleneck in robotic deployment, making it potentially valuable for real-world applications.
# Cons
* Lack of Empirical Evidence: The paper appears to be in a preprint stage and lacks real-world deployment data, leaving claims about real-world performance unverified.
* Limited Discussion on Practical Challenges: While the framework is theoretically sound, more discussion on potential implementation challenges (e.g., variability in real-world lighting and materials) would improve its applicability.
* Narrow Focus on RGB Images: Limiting the approach to RGB images may restrict applicability to specific scenarios or environments where other sensors (e.g., depth, tactile) could enhance robustness.
* Dependency on Simulation Quality: The success of the framework depends heavily on the accuracy of the simulation environment, which may limit generalizability across different robots and tasks.

---

### Official Review · ~Bryan_Constantine_Sadihin1 · 2024-11-09
**Review for "Bridging the Sim-to-Real Gap for Efficient and Robust  Robotic Skill Acquisition"**

**Rating:** 9
**Confidence:** 2

**Review:**

Strength:
1. A really detailed explanation for related research
2. A well-chosen topic, addressing the transfer challenge from simulation to real-world applications is a prospective step toward robotic autonomy.

Weakness:
1. There is no mention of evaluation metrics or baseline models for comparison, which is crucial for assessing research effectiveness.

---

### Official Review · ~Xun_Wang10 · 2024-11-10
**Review for "Bridging the Sim-to-Real Gap for Efficient and Robust Robotic Skill Acquisition"**

**Rating:** 9
**Confidence:** 4

**Review:**

This proposal presents a novel architecture combining Imitation Learning (IL) and Reinforcement Learning (RL) to address the sim-to-real gap in robotics. This architecture leverages the advantages of both learning paradigms, enabling efficient skill acquisition in simulators and robust deployment in real-world settings.

Strength: The proposal thoroughly elaborates on the importance of the sim-to-real gap issue in robotics and provides a relatively comprehensive literature review.

Weakness: In the methodology section, the description of how IL and RL are combined may lack sufficient detail. Additionally, there is little mention of how the proposed framework will be evaluated.

---

### Official Review · ~Chengming_Shi1 · 2024-11-11

**Rating:** 8
**Confidence:** 4

**Review:**

### Summary

The proposal "Bridging the Sim-to-Real Gap for Efficient and Robust Robotic Skill Acquisition" aims to develop a novel framework that combines Imitation Learning (IL) and Reinforcement Learning (RL) to enhance the transferability of robotic skills from simulation to real-world environments. The research focuses on addressing the sim-to-real gap by using RGB images and low-cost simulation data to train robots, with the goal of achieving high success rates in real-world tasks without the need for real-world data during training.

### Pros

1. **Combination of IL and RL**: The proposal leverages the strengths of both IL and RL, which could lead to a more efficient and robust learning paradigm for robotic skill acquisition.
2. **Addressing the Sim-to-Real Gap**: The focus on minimizing the sim-to-real gap is crucial for the broader application of simulation-trained models in real-world scenarios.
3. **Cost-Effectiveness**: By using simulation data, the proposed method could significantly reduce the costs associated with real-world data collection, including time, equipment, and potential robot damage.
4. **Innovation in Transferability**: The approach of translating real-world images into a simulation-style format could be a breakthrough in improving the transferability of learned policies.
5. **Direct Deployment**: The aim to develop policies that can be directly deployed in real-world environments without additional real-world training is a significant step towards more autonomous robotic systems.

### Cons

1. **Complexity of Integration**: Combining IL and RL in a way that effectively bridges the sim-to-real gap may be technically challenging and could require complex algorithmic solutions.
2. **Real-World Performance**: There is a risk that even with improved transferability, the performance in real-world scenarios may not meet the high standards set by the proposal.
3. **Data Translation Limitations**: The translation of real-world images into a simulation-style format may not fully capture the complexity and variability of real-world conditions.
4. **Dependence on Simulation Quality**: The success of the proposed method is highly dependent on the quality and realism of the simulation environment, which may not always be achievable.
5. **Scalability**: While the method aims to be cost-effective, there is a question of scalability and whether the approach can be generalized to a wide range of robotic tasks and environments.

---

### Official Review · ~Xuancheng_Li1 · 2024-11-12

**Rating:** 9
**Confidence:** 4

**Review:**

Summary
This paper proposes a framework that addresses the sim-to-real gap in robotic learning by integrating Imitation Learning (IL) and Reinforcement Learning (RL). The approach leverages simulated environments for efficient data generation while applying techniques to improve the transferability of learned behaviors to real-world conditions. By combining IL's rapid skill acquisition from expert demonstrations with RL's self-discovery capabilities, the framework seeks to enhance both the speed and robustness of robotic skill acquisition.

Strengths
The proposal provides a well-thought-out solution to a major challenge in robotics, demonstrating an innovative combination of IL and RL to balance data efficiency with learning autonomy. The approach holds promise for improving the effectiveness of simulation-based training in real-world applications, potentially accelerating the development of autonomous robotic systems.

Weaknesses
The proposal could benefit from more specifics on the techniques used to bridge the sim-to-real gap, such as domain adaptation or sensor calibration methods. Additionally, an outline of specific metrics for assessing transfer success in real-world tasks would provide clearer insights into the framework's practical effectiveness.

Conclusion
This framework offers a promising advancement in robotic skill acquisition, addressing the practical challenges of simulation-based training. With further refinement, particularly in sim-to-real transfer techniques, this approach could significantly contribute to the field of robotics by enabling more autonomous and adaptable robots.

---

### Official Review · ~Kittaphot_Saengprachathanarak1 · 2024-11-12
**Review of "Bridging the Sim-to-Real Gap for Efficient and Robust Robotic Skill Acquisition"**

**Rating:** 8
**Confidence:** 4

**Review:**

This proposal presents a strong and relevant approach to bridging the sim-to-real gap in robotic skill acquisition. By combining Imitation Learning (IL) and Reinforcement Learning (RL), it aims to create a framework that enhances the efficiency of skill acquisition while ensuring high transferability to real-world environments. The authors provide a detailed discussion of the limitations of existing methods, such as the high cost and risk of real-world data collection, and address the sim-to-real gap effectively by focusing on domain adaptation strategies. The proposed use of standardized simulation-style translation for real-world images to maximize consistency is innovative and has the potential to improve real-world task performance significantly. However, the proposal could benefit from more specifics on evaluation metrics and the anticipated challenges in policy robustness when using only simulated RGB data. Overall, this research has promise for advancing autonomous robotic capabilities.